# Use of SSR Markers for the Exploration of Genetic Diversity and DNA Finger-Printing in Early-Maturing Upland Cotton (*Gossypium hirsutum* L.) for Future Breeding Program

Zhengcheng Kuang [1,†], Caisheng Xiao [1,†], Muhammad Kashif Ilyas [2,†], Danish Ibrar [2], Shahbaz Khan [2,*], Lishuang Guo [1], Wei Wang [3], Baohua Wang [4], Hui Huang [1], Yujun Li [1], Yuqiang Li [1], Juyun Zheng [5], Salman Saleem [2], Ayesha Tahir [6], Abdul Ghafoor [2] and Haodong Chen [1,*]

1. Hunan Institute of Cotton Sciences Research, National Hybrid Cotton Research Promotion Center, Changde 415101, China; 18873692752@163.com (Z.K.); xiaoheziyi11@sina.com (C.X.); mksgls@163.com (L.G.); huangh93@126.com (H.H.); 15974267467@163.com (Y.L.); lizhong863@sina.com (Y.L.)
2. National Agricultural Research Centre, Park Road, Islamabad 45500, Pakistan; kashifnarc@gmail.com (M.K.I.); danish.uaar@gmail.com (D.I.); salman146@yahoo.com (S.S.); ghafoor59pk@yahoo.com (A.G.)
3. Agricultural Sciences Institute of Coastal Area of Jiangsu, Yancheng 224002, China; ww462@126.com
4. School of Life Sciences, Nantong University, Nantong 226019, China; bhwang@ntu.edu.cn
5. Economic Crop Research Institute, Xinjiang Academy of Agricultural Sciences, Urumqi 830091, China; zjypp8866@126.com
6. Department of Biosciences, COMSATS University, Chak Shahzad Campus, Park Road, Islamabad 45550, Pakistan; ayesha.tahir@comsats.edu.pk
* Correspondence: shahbaz2255@gmail.com (S.K.); chdmks@163.com (H.C.)
† These authors contributed equally to this work.

**Abstract:** DNA fingerprinting and genetic diversity analysis of 79 early-maturing upland cotton (*Gossypium hirsutum* L.) cultivars were performed using Simple Sequence Repeat (SSR) molecular markers. Out of 126 pairs of SSR primers, we selected 71 pairs that gave good polymorphisms and clear bands, had good stability, and showed even distribution on the cotton chromosomes, and 142 polymorphic genotypes were amplified. The average number of alleles amplified with the SSR primers was 2.01. The polymorphism information content (PIC) of the markers ranged from 0.1841 to 0.9043, with an average of 0.6494. The results of fingerprint analysis showed that nine varieties had characteristic bands, and at least six primer pairs could be used to completely distinguish all 79 cotton accessions. Using NTSYS-pc 2.11 cluster analysis, the genetic similarity coefficients between the cotton genotypes ranged from 0.3310 to 0.8705, with an average of 0.5861. All cotton accessions were grouped into five categories at a similarity coefficient of 0.57, which was consistent with the pedigree sources. At the same time, the average genetic similarity coefficients of early-maturing upland cotton varieties in China showed a low-high-low pattern of variation over time, revealing the development history of early-maturing upland cotton varieties from the 1980s to the present. This also indirectly reflects that in recent years, China's cotton breeders have focused on innovation and have continuously broadened the genetic resources for early-maturing upland cotton.

**Keywords:** early-maturing upland cotton; DNA fingerprinting; genetic diversity; molecular markers

## 1. Introduction

Cotton is an important economic crop in China and provides the leading raw material used in the textile industry [1–3]. In recent years, cotton cultivation in China has shown a trend of moving eastward, westward, and northward [4]. In order to ensure the safety of cotton production and utilization, China will continue to maintain the three existing cotton-producing regions; the Northwest inland (Xinjiang), the Yellow River Basin (YRB), and the Yangtze River Basin (YTRB) for the foreseeable future. However, expansion of cotton production in Xinjiang is limited due to water shortages. The cotton plantation

area in YTRB will be maintained at 660,000 hm$^2$ well into the future because of its suitable geographical climate, developed cotton spinning industry, and stable market demand. Cotton is the dominant crop in the YTRB, although its competitiveness is weak due to its longer growth period, high labor requirements, and high production cost. Therefore, it is urgent to select and breed new cotton varieties suitable for mechanized production in order to reduce labor costs and increase cotton planting efficiency [5].

Early-maturing upland cotton has the typical characteristics of a relatively short growth period, and concentrated flowering and boll opening [6,7]. The development of early-maturing upland cotton is one of the main targets of cotton breeding in the YTRB for the future; this will not only allow for two crops per year by rotation with winter crops such as wheat and rape, but is also suitable for mechanized harvesting to achieve simple and efficient cotton production [8]. A total of 79 early-maturing cotton accessions were collected and introduced from northern China to improve the local core germplasm resources that have long growth periods. To fully realize the genetic variation present in the introduced germplasm resources, it is necessary to study the genetic diversity of the 79 accessions.

Molecular marker technology is one of the main tools used for studying the genetic diversity of cotton varieties, both in China and overseas, and the marker types include, but are not limited to, restriction fragment length polymorphisms (RFLPs), random amplified polymorphic DNA (RAPD), amplified fragment length polymorphisms (AFLPs), simple sequence repeats (SSRs), and single nucleotide polymorphisms (SNPs) [9–11]. Of these marker types, SSRs have the advantages of high polymorphism, good reproducibility, co-dominance, and simple operation [12,13]. SSRs have been effectively utilized for cotton DNA fingerprinting, genetic diversity analysis, and QTL mapping [8,14–18], and this has enhanced the protection of cotton germplasm resources and enabled the genetic improvement of cotton varieties in China. DNA fingerprinting through PCR-based molecular markers had been reported in cotton. The clustering of cotton genotypes based on lint color (white or colored) was performed using RAPD markers, and a distinct grouping of colored- and white-linted genotypes was observed [19]. Twenty SSR markers were employed for the DNA fingerprinting of 30 major upland cotton cultivars. Four primer pairs were specific to four cotton cultivars, while 26 cotton cultivars can be distinguished by using two primer pairs [20]. It is also evident that SSR markers are not related to any agronomic traits of cotton plant, hence the ability of such SSR primers to distinguish cotton cultivars, specifically some new transgenic lines, was very limited [21]. Han et al. [22] used SSR markers to construct fingerprints and analyze the genetic diversity of 27 cotton accessions from 2009 to 2010. Li et al. [23] used 20 pairs of SSR primers to construct fingerprints of the BaiMian cotton series, while Kuang et al. [24] found a correlation between cultivars and geographical origins by analyzing the genetic diversity in the main cotton varieties in China using SSR markers. In the current research, DNA fingerprinting, genetic diversity, and the genetic relationships among 79 early-maturing upland cotton accessions were analyzed using SSR markers. The genotyping of the 79 early-maturing upland cotton genotypes will be helpful in developing a database of high throughput SSR-based marker detection systems, which in future could be utilized for a large-scale database of DNA fingerprints for the identification and authentication of cotton cultivars. The results of our study will also provide genetic resources for the breeding of new varieties of early-maturing upland cotton in Hunan and the YTRB through a systematic understanding of the genetic backgrounds of 79 early-maturing upland cotton germplasm accessions.

## 2. Materials and Methods

### 2.1. Experimental Material

Seventy-nine early-maturing upland cotton accessions were acquired from the National Cotton Mid-term Gene Bank (Anyang, Henan, China). These accessions were collected from six regions, including the Institute of Cotton Research of the Chinese Academy of Agricultural Sciences (ICR-CAAS), which is considered as a separate branch because it is a national institute and its cotton varieties are sui generis (of their own kind), the

YRB (Henan, Shanxi, Shandong, Jiangsu), the Northwest Inland Region (Xinjiang, Gansu), the Liaohe River Basin, the United States, and the former Soviet Union (Table S1). All materials were planted in 2015 at the Deshan Experimental Field of the Hunan Institute of Cotton Sciences Research. Polymorphic SSR markers were selected for testing because they have been used on other materials for many years, and the primers were synthesized by Shanghai Yingjun Biotechnology Co., Ltd. (Guangzhou, China). PCR reagents (Taq DNA polymerase, dNTPs, DNA marker size standard) of Beijing Quanjin Biotechnology Co., Ltd. (Beijing, China) were utilized. In this experiment, no endangered or protected species was used.

### 2.2. DNA Extraction, PCR Amplification and Electrophoresis Detection

Young leaves collected from the field were flash frozen in liquid nitrogen and grounded to powder form. Total genomic DNA was extracted by the CTAB DNA extraction procedure, as described by Zhang [25], with some modifications. The DNA was quantified by UV Mass spectrophotometer at A260 and A280 optical density. To check the quality and quantity of isolated genomic DNA by electrophoresis, the DNA was loaded in agarose gel (1.5%), along with the standard DNA, and the final concentrations were adjusted to 50 ng $\mu L^{-1}$ and stored at $-20\,°C$. The PCR system consisted of 10 $\mu L$ reactions containing 1 $\mu L$ $10\times$ reaction buffer (including 10 mmol $L^{-1}$ $MgCl_2$), 0.5 $\mu L$ dNTPs (10 mmol $L^{-1}$ of each), 0.4 $\mu L$ of the forward and reverse primers (10 $\mu mol\ L^{-1}$), 0.1 $\mu L$ Taq DNA polymerase (5 U $\mu L^{-1}$), 0.5 $\mu L$ cotton DNA (50 ng $\mu L^{-1}$), and 7.1 $\mu L$ $ddH_2O$. The PCR amplification protocol was as follows: pre-denaturation at 95 $°C$ for 30 min, followed by 30 cycles of denaturation at 94 $°C$ for 45 s, annealing at 59 $°C$ for 45 s, and extension at 72 $°C$ for 1 min, with a final extension at 72 $°C$ for 3 min. Amplification reactions were stored at 4 $°C$. The PCR products were separated by polyacrylamide gel electrophoresis (PAGE) on 8% gels. PAGE gel mixture was prepared by mixing 8% PAGE gel with 20 $\mu L$ of tetramethylethylenediamine (TEMED) and 200 $\mu L$ fresh 10% ammonium per sulphate (APS). Gel solutions was then poured into 1.5 mm thickness spacer plates. Electrophoresis was performed in TBE buffer at 200 V for 45 min, and the bands were observed by silver staining and photographed.

### 2.3. Band Recording and Data Analysis

DNA fragments amplified with a primer pair that had the same migration position in the PAGE gel were recorded as 1, absence of a band was recorded as 0, and bands that were blurred or had a deletion were recorded as 999, and the [0,1] binary data matrix was constructed. A 100 kb DNA ladder was also loaded to record the length of DNA fragments amplified by each SSR primer. The polymorphic information content (PIC) of the SSR primers was calculated as PIC $= 1 - \Sigma pi^2$, genotypic diversity was calculated as H′ $= -\Sigma pi$ ln$pi$, and the number of effective alleles per locus was Ne $= 1/(\Sigma Pi^2)$, where Pi represents the gene frequency of the ith allelic variation at a certain locus. Genetic analysis of the 79 cotton accessions was performed using NTSYS-pc 2.1 software. The Jaccard similarity coefficient was found using the Qualitative program in the Similarity for the original [0,1] binary data matrix obtained from the EST-SSR markers. Based on the genetic similarity coefficient, the UPGMA (unweighted pair group method with arithmetic mean) algorithm in the SAHN program was used for cluster analysis, and the phenogram was generated using the Treeplot module under Graphics.

### 3. Results

#### 3.1. Selection of SSR Primers and Polymorphisms in the Amplified Products

A total of 126 pairs of SSR primers were selected by using eight samples of DNA from ICR-CAAS14 and 'Xinluzao6' as templates (Figure 1), and 71 pairs of SSR primers that gave good polymorphisms, clear amplified bands, and excellent stability were selected for further study. Except for the unknown chromosome information for 17 pairs of primers, the other primers covered all cotton chromosomes except for Chr.04, Chr.08, Chr.16, and Chr.26.

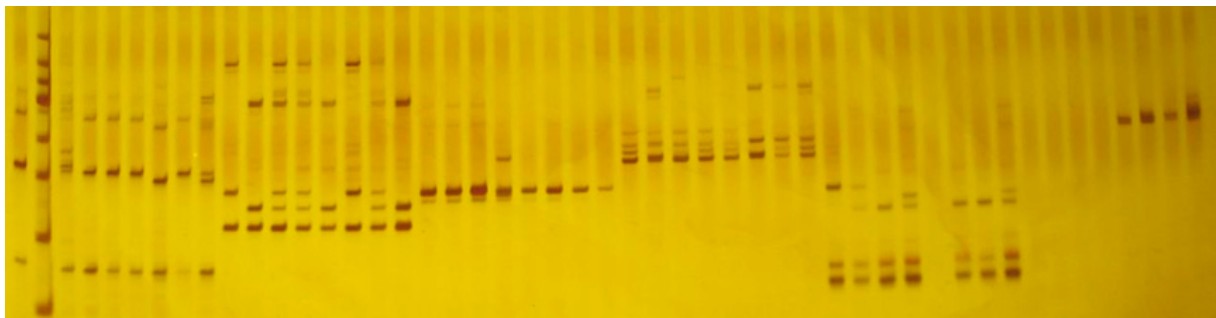

**Figure 1.** A silver-stained gel showing primer screening for polymorphic SSRs.

A total of 142 effective allelic variants were detected by amplifying DNA from the 79 accessions with the selected 71 pairs of SSR primers. The average number of alleles detected for each SSR primer pair was 2.01, with a range of 1 to 6. The effective allele numbers ranged from 1.2256 to 10.4502, with an average of 3.4379. Polymorphism information content (PIC) ranged from 0.1841 to 0.9043, with an average of 0.6494. The PIC of the primer pair MON_CGR5565 had the highest value of 0.9043, and the primer pair NAU4044 had the second highest PIC value of 0.8822, whereas the primer NAU3181 had the lowest PIC value of 0.1841. Genetic diversity (H′) ranged from 1.2256 to 10.4502 (Table 1).

**Table 1.** SSR marker loci, chromosomal locations, and SSR-PCR polymorphism data.

| SN | SRR Locus | Chromosome | PIC | Genetic Diversity (H′) | Effective Number of Alleles (Ne) |
|----|-----------|------------|-----|------------------------|----------------------------------|
| 1 | NAU4073 | Chr.01 | 0.8168 | 5.7030 | 5.4582 |
| 2 | NAU2457 | Chr.01 | 0.2577 | 1.5311 | 1.3471 |
| 3 | NAU2083 | Chr.01 | 0.4356 | 1.8725 | 1.7717 |
| 4 | NAU4044 | Chr.01 | 0.8822 | 9.1008 | 8.4884 |
| 5 | MON_COT064 | Chr.02 | 0.4980 | 1.9960 | 1.9920 |
| 6 | NAU1190 | Chr.03 | 0.7956 | 5.3171 | 4.8933 |
| 7 | MON_CGR6528 | Chr.03 | 0.7423 | 3.9381 | 3.8800 |
| 8 | MON_CGR6683 | Chr.03 | 0.4576 | 1.9158 | 1.8437 |
| 9 | NAU1269 | Chr.05 | 0.7314 | 3.8504 | 3.7234 |
| 10 | MON_CGR5732 | Chr.05 | 0.7307 | 3.8447 | 3.7138 |
| 11 | NAU1225 | Chr.05 | 0.7307 | 3.8447 | 3.7138 |
| 12 | MON_DC40122 | Chr.05 | 0.6884 | 3.5157 | 3.2096 |
| 13 | NAU1221 | Chr.05 | 0.7307 | 3.8447 | 3.7138 |
| 14 | MON_CGR5651 | Chr.06 | 0.6811 | 3.4515 | 3.1354 |
| 15 | MON_DPL0702 | Chr.06 | 0.7349 | 3.8787 | 3.7721 |
| 16 | MON_COT002 | Chr.06 | 0.7266 | 3.8116 | 3.6572 |
| 17 | BNL1694 | Chr.07 | 0.7442 | 3.9534 | 3.9092 |
| 18 | MUSS095 | Chr.07 | 0.7431 | 3.9447 | 3.8921 |
| 19 | MON_DC30218 | Chr.07 | 0.5000 | 2.0000 | 2.0000 |
| 20 | NAU3859 | Chr.09 | 0.4980 | 1.9960 | 1.9920 |
| 21 | DPL0431 | Chr.10 | 0.6873 | 3.4966 | 3.1981 |
| 22 | NAU3784 | Chr.11 | 0.7071 | 3.6561 | 3.4141 |
| 23 | MON_CER0098 | Chr.11 | 0.7250 | 3.8002 | 3.6363 |
| 24 | NAU3563 | Chr.11 | 0.8579 | 7.4448 | 7.0361 |
| 25 | NAU2671 | Chr.12 | 0.6978 | 3.5853 | 3.3095 |
| 26 | MON_DPL0491 | Chr.12 | 0.2397 | 1.4972 | 1.3153 |
| 27 | NAU3991 | Chr.13 | 0.6801 | 3.4455 | 3.1264 |
| 28 | MON_COT009 | Chr.13 | 0.6886 | 3.5149 | 3.2116 |
| 29 | BNL1421 | Chr.13 | 0.7107 | 3.6867 | 3.4567 |
| 30 | NAU3308 | Chr.14 | 0.6721 | 3.3728 | 3.0500 |
| 31 | GH304 | Chr.15 | 0.7085 | 3.6704 | 3.4305 |

**Table 1.** *Cont.*

| SN | SRR Locus | Chromosome | PIC | Genetic Diversity (H′) | Effective Number of Alleles (Ne) |
|----|-----------|------------|-----|------------------------|----------------------------------|
| 32 | MUSS440 | Chr.15 | 0.8784 | 8.9176 | 8.2218 |
| 33 | NAU2343 | Chr.15 | 0.7354 | 3.8824 | 3.7790 |
| 34 | BNL2646 | Chr.15 | 0.6360 | 3.1112 | 2.7475 |
| 35 | NAU2742 | Chr.17 | 0.6635 | 3.3170 | 2.9714 |
| 36 | MON_DPL0308 | Chr.18 | 0.5896 | 2.7536 | 2.4365 |
| 37 | NAU3011 | Chr.18 | 0.4980 | 1.9960 | 1.9920 |
| 38 | NAU5262 | Chr.18 | 0.7407 | 3.9252 | 3.8560 |
| 39 | MON_CGR6151 | Chr.19 | 0.4768 | 1.9539 | 1.9115 |
| 40 | NAU1187 | Chr.19 | 0.7307 | 3.8447 | 3.7138 |
| 41 | NAU1042 | Chr.19 | 0.7307 | 3.8447 | 3.7138 |
| 42 | MON_CGR5590 | Chr.19 | 0.7193 | 3.7542 | 3.5631 |
| 43 | TMB1791 | Chr.19 | 0.7387 | 3.9101 | 3.8277 |
| 44 | MON_CGR6439 | Chr.20 | 0.2604 | 1.5362 | 1.3520 |
| 45 | DPL0442 | Chr.20 | 0.7317 | 3.8542 | 3.7271 |
| 46 | MON_SHIN1421 | Chr.20 | 0.7418 | 3.9344 | 3.8728 |
| 47 | MON_CGR5565 | Chr.20 | 0.9043 | 11.0904 | 10.4502 |
| 48 | BNL1551 | Chr.21 | 0.7006 | 3.6046 | 3.3401 |
| 49 | GH222 | Chr.22 | 0.8094 | 5.5713 | 5.2460 |
| 50 | MON_CGR6410 | Chr.22 | 0.7378 | 3.9022 | 3.8136 |
| 51 | CIR253 | Chr.22 | 0.6158 | 2.9431 | 2.6031 |
| 52 | MUSS139 | Chr.23 | 0.7426 | 3.9408 | 3.8848 |
| 53 | MON_CGR5202 | Chr.24 | 0.3230 | 1.6552 | 1.4772 |
| 54 | MON_CGR6932 | Chr.25 | 0.7423 | 3.9381 | 3.8800 |
| 55 | CRI151 | | 0.7312 | 3.8485 | 3.7200 |
| 56 | MON_CGR6389 | | 0.7430 | 3.9442 | 3.8913 |
| 57 | NAU3181 | | 0.1841 | 1.3919 | 1.2256 |
| 58 | MON_C2-0118 | | 0.7397 | 3.9179 | 3.8418 |
| 59 | CRI002 | | 0.6296 | 3.0356 | 2.7000 |
| 60 | MUCS375 | | 0.3519 | 1.7103 | 1.5429 |
| 61 | MON_CGR6784 | | 0.6384 | 3.1230 | 2.7658 |
| 62 | CIR096 | | 0.7210 | 3.7659 | 3.5837 |
| 63 | NAU3254 | | 0.8289 | 6.4664 | 5.8436 |
| 64 | MON_DPL0133 | | 0.5479 | 2.4396 | 2.2119 |
| 65 | MON_CER0168 | | 0.7397 | 3.9179 | 3.8421 |
| 66 | MUSB0175 | | 0.4734 | 1.9470 | 1.8989 |
| 67 | MON_DPL0906 | | 0.4999 | 1.9998 | 1.9997 |
| 68 | GH111 | | 0.4734 | 1.9470 | 1.8989 |
| 69 | GH112 | | 0.4647 | 1.9297 | 1.8680 |
| 70 | MON_DC40266 | | 0.7037 | 3.6271 | 3.3744 |
| 71 | MON_DC40286 | | 0.6888 | 3.5120 | 3.2131 |

### 3.2. DNA Fingerprinting Analysis of the 79 Cotton Accessions

Fingerprinting analysis of the 79 early-maturing upland cotton accessions was performed using 71 pairs of SSR primers. It was found that nine accessions had characteristic bands for which only one primer pair was needed to distinguish each accession from others. Among them, ICR-CAAS64 had two characteristic primers, 'Xinluzao20', 'Xinluzao25', 'Jiumian9', 'Liaomian5', 'Liaomian17', 'Liaomian19', 'Lumianyan28', and 'Jinmian23', and each had one characteristic primer (Table 2). The primer pair NAU4044 was able to uniquely identify four varieties including 'Xinluzao25', 'Jiumian9', 'Lumianyan28', and 'Liaomian 5', while primer NAU3254 can distinguish three varieties, i.e., 'Xinluzao20', 'Liaomian17', and 'Liaomian19'. These results indicated that these two primer pairs had abundant polymorphism, strong discrimination power, and numerous characteristic bands, and could be used as preferred markers in the identification of fingerprints.

**Table 2.** Cotton accessions identified with specific SSR primer pairs.

| Cultivar | Specific Primer | Cultivar | Specific Primer |
|---|---|---|---|
| ICR-CAAS64 | NAU1190, MUSS440 | Xinluzao20 | NAU3254 |
| Liaomian17 | NAU3254 | Liaomian19 | NAU3254 |
| Xinluzao25 | NAU4044 | Jiumian9 | NAU4044 |
| Liaomian5 | NAU4044 | Lumianyan28 | NAU4044 |
| Jinmian23 | NAU4073 | | |

A total of 55 of the 79 cotton accessions could be identified by three pairs of primers, NAU4044, MUSS440, and MON_CGR5565, also having high PIC values, strong discriminative power, clear bands on the gels, and high reproducibility. Seventy-two varieties could be identified by adding another primer pair, GH222. While using a combination of NAU4044, MUSS440, MON_CGR5565, GH222, NAU1190, and BNL1694, all of the 79 upland cotton varieties could be completely distinguished from one another (Table S2).

*3.3. Genetics Diversity Analysis*

Similarity coefficients between varieties were calculated using NTSYS-pc 2.11 and Microsoft Excel software. Results showed that the genetic similarity coefficients among the 79 early-maturing upland cotton accessions ranged from 0.3310 ('Jinmian36' and 'Xinluzao25') to 0.8705 ('Liaomian15' and 'Liaomian18'), with an average of 0.5861. The similarity coefficients between varieties that were <0.4 (large genetic difference) accounted for 1.1%, while 11.9% of the varieties exhibited genetic similarity coefficients between 0.4 and 0.5: 44.1% at 0.5–0.6, 35.2% at 0.6–0.7, and 7.1% at 0.7–0.8. Similarity coefficients >0.8 accounted for 0.5% (Figure 2). These results showed that 86.9% (44.1 + 35.2 + 7.1 + 0.5) of the cotton varieties studied had a similarity index greater than 0.5. Although most of the genotypes studied are genetically closer to each other, a fair amount of genetic diversity still existed.

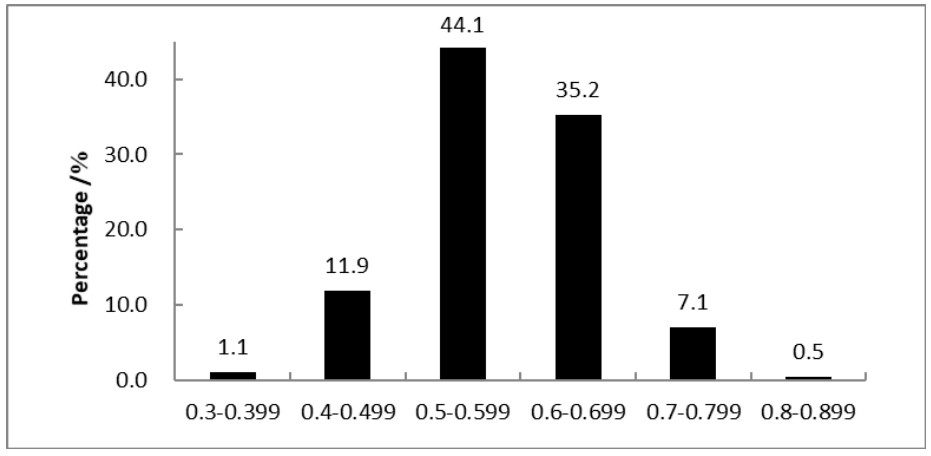

**Figure 2.** Similarity coefficients for the 79 early-maturing upland cotton accessions.

*3.4. Region-Wise Genetic Diversity Analysis*

Seventy-nine early-maturing upland cotton accessions acquired from six different ecological regions that included the China Cotton Institute, the YRB (Henan, Shanxi, Shandong, Jiangsu), the Northwest Inland Cotton Region (Xinjiang, Gansu), Liaohe River Basin, United States, and the former Soviet Union were analyzed region-wise for genetic diversity. By comparison, the accessions from the former Soviet Union had the smallest average genetic similarity coefficients of the six regions, which increased in the order of the YRB cotton area, the United States, the China Cotton Institute, the Liaohe River Basin Early-maturing Cotton Area, and the Northwest Inland Cotton Area, indicating that there is ample genetic diversity in cotton resources imported from abroad. At the same time, the

genetic diversity of accessions from the YRB cotton region is relatively high, which may be related to the complex geographical diversity of the YRB cotton region and the dispersion of breeders in Henan, Shanxi, Shandong, Jiangsu, and other provinces.

Genetic similarity coefficients of accessions from the six different regions were between 0.5575 and 0.6143, and the highest similarity coefficients were found between accessions from China and the USA. This indicates that early-maturing upland cotton varieties selected by ICR-CAAS have close genetic relationships to selections from the USA. In general, the Chinese cotton germplasm is more frequently exchanged for resources from the USA compared with other regions. The lowest genetic similarity coefficient in the early-maturing cotton areas is between the YRB and the Liaohe River Basin, with a value of 0.5575, and the second lowest is between the YRB and the Northwest Inland Cotton Area, with a value of 0.5636 (Table 3). The underlying reason for this may be that the YRB cotton-growing area has a better climate with warmer conditions, resulting in more varieties of early-maturing upland cotton and larger differences between varieties than the early-maturing cotton areas of the Liaohe River Basin and the Northwest Inland Cotton Area.

**Table 3.** Genetic diversity of cotton cultivars from the six cotton-growing regions in China.

| Region | Genetic Similarity Coefficient | CAAS | YRB | Northwest Inland Region | Liaohe River Basin | United States | Former Soviet Union |
|---|---|---|---|---|---|---|---|
| CAAS | Max | 0.8058 | 0.7817 | 0.8169 | 0.8451 | 0.7447 | 0.6691 |
|  | Min | 0.3630 | 0.3643 | 0.3704 | 0.3582 | 0.4789 | 0.4296 |
|  | Mean | 0.6032 | 0.5796 | 0.5943 | 0.5824 | 0.6143 | 0.5672 |
| YRB | Max |  | 0.7899 | 0.8028 | 0.7606 | 0.7042 | 0.7042 |
|  | Min |  | 0.4085 | 0.3310 | 0.3475 | 0.3521 | 0.4366 |
|  | Mean |  | 0.5802 | 0.5636 | 0.5575 | 0.5911 | 0.5661 |
| Northwest Inland Region | Max |  |  | 0.8239 | 0.8310 | 0.7324 | 0.7254 |
|  | Min |  |  | 0.4225 | 0.3582 | 0.4296 | 0.4296 |
|  | Mean |  |  | 0.6221 | 0.5936 | 0.5799 | 0.5894 |
| Liaohe River Basin | Max |  |  |  | 0.8705 | 0.7465 | 0.6812 |
|  | Min |  |  |  | 0.3944 | 0.4718 | 0.4014 |
|  | Mean |  |  |  | 0.6134 | 0.5783 | 0.5684 |
| United States | Max |  |  |  |  | 0.5845 | 0.6549 |
|  | Min |  |  |  |  | 0.5845 | 0.5357 |
|  | Mean |  |  |  |  | 0.5845 | 0.6027 |
| Former Soviet Union | Max |  |  |  |  |  | 0.5286 |
|  | Min |  |  |  |  |  | 0.5286 |
|  | Mean |  |  |  |  |  | 0.5286 |

Comparisons of the genetic similarity coefficients between domestic and foreign early-maturing upland cotton varieties showed that, except for the Northwestern Inland Cotton Area, the genetic similarity coefficients between cotton varieties from the ICR-CAAS, the YRB, and the Liaohe River Basin are higher. This suggests that early-maturing upland cotton grown in the ICR-CAAS, the YRB, and the Liaohe River Basin in the early-maturing cotton area contains more American germplasm. Because of the introduction and utilization of early-maturing upland cotton in China, the majority of early-maturity genetic resources came from American gold-colored cotton. The genetic similarity coefficients between accessions from the Northwestern Inland Cotton Region and the former Soviet Union are relatively high. This may be because the Northwest Inland Cotton Region is adjacent to the former Soviet Union, so it is easier to introduce germplasm resources into Northern China from there.

The results of our study showed that the average genetic similarity coefficient of bred and certified varieties with early maturity collected from different areas in China had a low-high-low pattern of variation. 'Jinzhong200', 'Xinluzao1', 'Lumian1', 'Heishanmian1', and 'Liaomia5', which were selected prior to the 1980s, had the lowest genetic similarity coefficient of 0.5704. Nine varieties 'ICR-CAAS10', 'ICR-CAAS14', 'Xinluzao3', 'Liaomian6', 'Liaomian7, 'Liaomian9', 'Sumian1', 'Yumian3', and 'Yumian5', which were selected in

the 1980s, have the highest average genetic similarity coefficient of 0.6306. Twenty-nine varieties selected in the 1990s, which include 'ICR-CAAS16', 'ICR-CAAS18', 'Xinluzao4', 'Liaomian10', 'Lumian10', 'Yumian7', 'Sumian10', and 'Jinmian23', have an average genetic similarity coefficient of 0.5993. The average similarity coefficient of 32 varieties i.e., 'ICR-CAAS42', 'ICR-CAAS50', 'Xinluzao13', 'Jiumian2', 'Liaomian17', 'Jinmian34', and 'Lumianyan27' selected after 2000 is 0.5791.

The average genetic similarity coefficients of early-maturing upland cotton varieties in China have shown a low-high-low pattern over time (Figure 3). This may be because, before the 1980s, domestic early-maturing upland cotton breeding was mainly carried out by introducing different early-maturing varieties from abroad and systematically using them in breeding. Since the early 1980s, cotton production and the cotton spinning industry have developed rapidly. Due to economic reform and the opening up of the country, transportation is more convenient, and the exchange of germplasm resources between breeding units has become frequent. In particular, a number of outstanding varieties (lines) such as 'Heishanmian1' and 'ICR-CAAS10' stand out from the competition and are used by other breeders as donor parents. This has resulted in closer genetic relationships between the varieties selected at this time, with higher genetic similarity coefficients and less genetic difference. In the 1990s, the difficulties of domestic distant hybridization were continuously overcome, and breeders consciously chose parental materials with complex genetic backgrounds for cross-breeding, which resulted in a significant reduction in the genetic similarity coefficients of cotton varieties and increased the genetic difference. After 2000, the use of modern breeding technologies (transgenics and molecular marker-assisted breeding) not only accelerated the cotton breeding process, but also broadened the source of available cotton genes, resulting in further reductions in the genetic similarity coefficients among new varieties of early-maturing upland cotton in China [26].

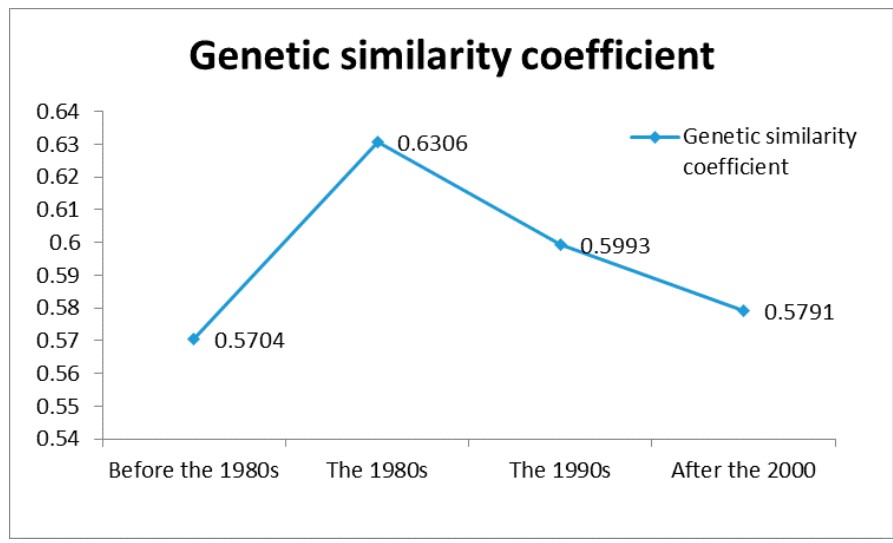

**Figure 3.** The genetic similarity coefficients of early-maturing upland cotton varieties developed over the past four decades.

### 3.5. Hierarchical Cluster Analysis

Based on the Jaccard similarity coefficient, 79 early-maturing upland cotton varieties were grouped using a hierarchical-based clustering method, i.e., UPGMA (Figure 4). The results showed that, at a genetic similarity coefficient of 0.87, all 79 early-maturing upland cotton varieties were completely separated, and at a coefficient of 0.57, all the 79 cotton varieties could be divided into five sub-groups or classes. Class I contains 42 varieties, including seven varieties from the China Cotton Institute, 20 from the Northwest Inland Cotton Area, 9 from the special early-maturing cotton area of the Liaohe River Basin, 4 varieties from the YRB cotton area, and one each from USA and former Soviet Union.

Class II contains 27 varieties, including 6 varieties from the China Cotton Institute, 6 from the Northwest Inland Cotton Area, 2 from the special early-maturing cotton area of the Liaohe River Basin, 11 from the YRB cotton area, and one variety each from the United States and former Soviet Union. Class III has only one variety, which is from the Northwestern Inland Cotton Area. Class IV also consists of a single variety of YRB, and Class V contains seven varieties, including 2 from Zhongmian, 2 varieties from the Liaohe Basin early-maturing cotton area, and 3 from the YRB cotton area. Among these five classes, most of the selected varieties of cotton grown in China clustered in Class I and Class II, accounting for 46.7% and 40.0%, respectively. Most of the varieties from the Northwestern Inland cotton area are in Class I, accounting for 71.4%, whereas most of the varieties from the Liaohe Basin are concentrated in Class I, accounting for 69.2%. Most of the cotton varieties from the YRB are concentrated in Class II, accounting for 57.9%. This shows that the clustering results reflect certain geographical distribution characteristics, and the genetic differences of the cultivars from the same area are relatively small, which is why they clustered together.

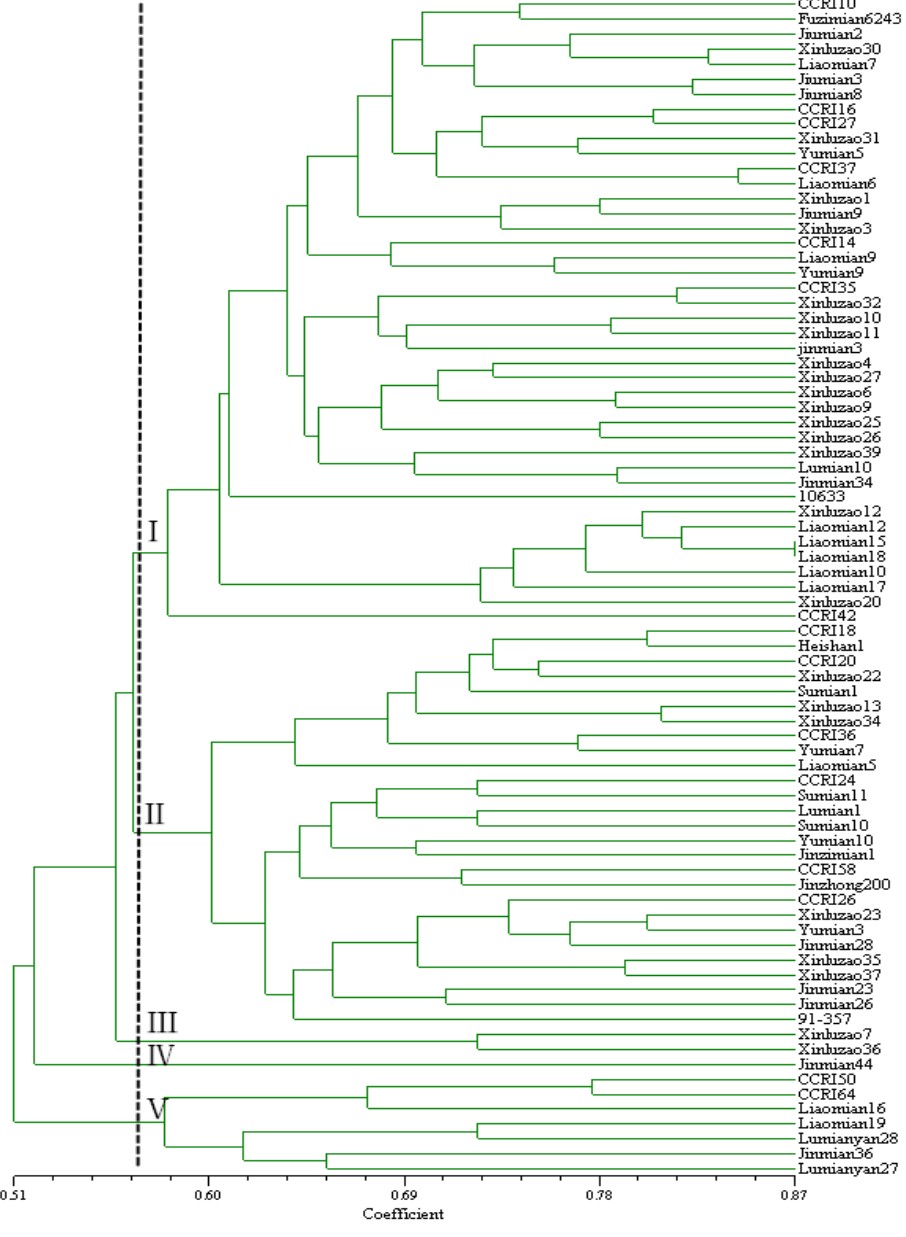

**Figure 4.** The genetic lineages of the 72 early-maturing upland cotton varieties based on UPGMA clustering.

*3.6. Pedigree/Parentage Analysis*

Seventy-two of the main varieties of early-maturing cotton used in this experiment can be divided into five classes (I, II, III, IV, and V) at a genetic similarity coefficient of 0.57, and this clustering pattern of the early-maturing upland cotton accessions was also consistent with their pedigree (Figure 5). For example, 'Liaomian15' and 'Liaomian18' were first clustered together, traced back to their pedigree sources, and were both found to contain 'Liao 1038' in their ancestry. 'Xinluzao6', 'Xinluzao9', 'Xinluzao27', 'Xinluzao31', and 'Xinluzao39' are grouped together in Class I, and their pedigree sources show that they are all descended from 'Bell Snow'. 'Xinluzao6', 'Xinluzao9', and 'Xinluzao27', clustered into a group with a genetic similarity coefficient of 0.87. We also identified several cases of incomplete matching. For example, 'Jiu Mian2' is in the 'ICR-CAAS16' lineage; 'Jinmian3' is in the 'Sumian1' lineage; seven varieties, including 'ICR-CAAS16', 'Zhongmian24', 'ICR-CAAS26', 'Yumian5', 'Yumian7', 'Yumian9', and 'Lumian10', are in the 'Zhongmian10' lineage; three varieties, including 'ICR-CAAS10', 'ICR-CAAS18', and 'Sumian 1', are in the 'Heishan Mian1' lineage; and three varieties, including 'Heishan1', 'Jinzhong200', and 'Jinzi 1', descend from the gold-line cotton pedigree. The clustering results indicates that seven varieties, including 'ICR-CAAS10', 'Jiu Mian2', 'ICR-CAAS16', 'Yumian5', 'Yumian9', 'Jinmian3', and 'Lumian10', are group together in Class I. Eight varieties, including 'Sumian1', 'Zhongmian24', 'Zhongmian26', 'Yumian7', 'ICR-CAAS18', 'Heishan Mian1', 'Jinzhong200', and 'Kings improved1', are in Class II.

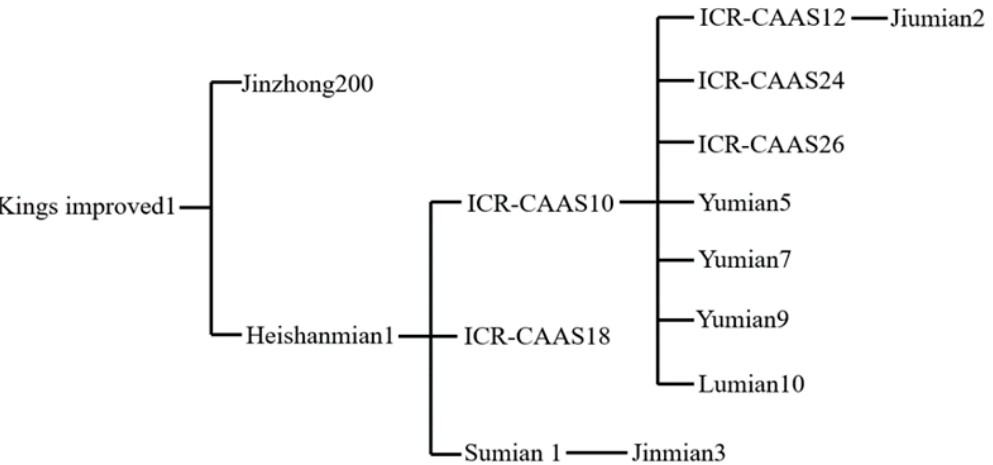

**Figure 5.** The pedigrees of 15 upland cotton varieties.

In addition, the genetic relationship of 'ICR-CAAS37' and 'Liaomian6' seems to be relatively distant, but they can be grouped together. However, the genetic relationship of 'Liaomian6' and 'Liaomian16' seems to be relatively close, but they do not cluster together. This suggests that the classification is not entirely dependent on pedigree but is also influenced by some other factors such as selection method, breeding process, and target traits.

## 4. Discussion

The current study examined the genetic variability among 79 early-maturing upland cotton accessions collected from six different geographical regions i.e., the China Cotton Institute, the YRB (Henan, Shanxi, Shandong, Jiangsu), the Northwest Inland Cotton Region (Xinjiang, Gansu), the Liaohe River Basin, the United States, and the former Soviet Union. The study of genetic divergence of any plant breeding population is essential as it constitutes the backbone of any cultivar breeding and improvement program [27]. This helps in the development of cultivars/hybrids that are suitable for rapidly changing climate by induction of new alien genes [28–30]. Therefore, the presence of genetic diversity is a pre-requisite for the development of new and improved crop varieties/hybrids to meet our present and future requirements. In this study, 79 SSR markers were screened, out of which

71 were selected because of their high polymorphism, reproducibility, and suitability for the diversity paneling of 79 upland early-maturing cotton accessions. SSR/microsatellite markers are highly preferable because of their codominant inheritance nature, high informative power, and transferability, making them a markers of choice for plant germplasm improvement schemes [31,32].

For interspecific and intraspecific hybridization, the presence of genetic variability is of prime importance to cotton as well as other field crop breeders [33,34]. The assessment of genetic variability and DNA-based finger printing characterizes genotypes and helps to associates them into different heterotic groups for hybridization based crop improvement programs [35–37]. For variability studies and DNA fingerprinting, different kinds of molecular markers, such as RFLPs, RAPDs, AFLPs, ISSRs, and SSR, have been utilized in cotton [10,35–40]. These studies were important for the identification and purity detection of cotton varieties in China, as well as for understanding the genetic relationships and origins of cotton varieties at the molecular level, which provided a theoretical basis for the rational selection of hybrid parents and the breeding of new cotton cultivars. These studies and results of the current experiments proved that the use of SSR makers is still effective in cotton for genotype characterization. However, there are fewer reports on the SSR-based genetic characterization and DNA fingerprinting of early-maturing upland cotton in China. In this study, DNA fingerprints of 79 early-maturing upland cotton accessions were constructed using 71 SSR markers, and the molecular data was analyzed to determine genetic similarities.

The average number of alleles observed in our study was 2.01, with a range of 1 to 6. However, the average PIC value observed was 0.6494, although the average number of alleles recorded in this experiment is slightly lower than many of the pervious findings of genetic dissimilarity in cotton. Zhu et al. [41] reported 6.02 alleles per locus in a study comprising of 557 *G. hirsutism* accessions. Javaid et al. [42] reported 3.72 alleles per locus in a study of genetic diversity in 22 cotton accessions using 30 SSR markers. Similarly, Gurmessa [43] reported 3.8 alleles per locus with 0.50 PIC value, whereas, according to our knowledge, only one study by McCarty et al. [44] reported a high number of alleles (7.9) per locus. Ali et al. [45] reported 6.3 and 0.73 as the average number of alleles and PIC value in cotton germplasm. However, the PIC average of the current study was quite high owing to the fairly large set of SSR markers utilized in this present investigation. Primer pair MON_CGR5565 showed the highest PIC value and average number of effective alleles. The number of alleles amplified by each marker not only corresponds to the diversity in the studied germplasm, but it also highly correlates with marker type, fragment separation technique applied, and the resolution [46].

Pedigree analysis can only reflect the relative genetic information between the varieties. Molecular marker analysis using marker loci distributed over the whole genome can more accurately reflect the genetic differences between varieties [47–50]. According to the hierarchical cluster diagram, all the 79 upland cotton accession can be grouped into five sub-classes at a similarity index of 0.57. This depicts a narrow genetic base in the studied material. In spite of narrow genetic diversity, cultivars for different geographical regions can be bred successfully [51,52]. Therefore, the present results could help the cotton breeders in China to identify suitable cultivars and engage them in appropriate breeding procedures for enhancing the cotton yield. Phylogenetic classification of upland cotton was also found to somewhat portray their geographical origin. However, region-wise genetic similarity index results showed that cotton accession from China and the USA are more closely related. This might be due to frequent germplasm exchange between these two locations. The findings of Çelİk [52] also correspond to the similar results of genetic diversity and clustering pattern in upland cotton, wherein a narrow genetic variability was observed in 28 cotton genotypes with 100 SSR markers. Past studies of genetic diversity in cotton accessions had divided them into two main groups and then into sub-groups, irrespective of their originating stations [53]. Buyyarapu et al. [54] also observed a phylogenetic tree

with four major sub-clusters for 23 species, while three species branched out individually in upland cotton genotypes.

The urge and speed of producing cotton cultivars with high productivity to increase profitability has decreased the genetic base, and this narrowing of the genetic base is increasing as a main hurdle in the development of successful cotton and other filed crop breeding programs [35,55]. If a breeder starts with a narrow/limited genetic base, the crop developed will be more susceptible to different biotic and abiotic stresses, with limited adaptability in the current scenarios of changing climatic patterns [52]. Therefore, efforts to broaden the genetic base of upland early-maturing cotton needs to be increased in order to have stable cotton production over the coming decades. Moreover, utilization of more advanced and reliable DNA identification techniques, such as high resolution melting (HRM), could be applied in conjunction with SSR-based primers for DNA fingerprinting. HRM has been proven to be highly effective in cultivar identification and the detection of single DNA inserts [56,57].

## 5. Conclusions

Genetic diversity studies can be effectively accomplished through SSR markers for the identification and selection of potential parents in order to achieve high production in early-maturing upland cotton. The current study found a narrow genetic base in 79 early-maturing cotton genotypes using 71 polymorphic, high reproducible SSR primers. The DNA fingerprinting information revealed by the SSR primers depicted that by using a combination of six primer pairs, i.e., NAU4044, MUSS440, MON_CGR5565, GH222, NAU1190, and BNL1694, all of the 79 cotton genotypes could be distinguished from each other. The present study revealed that the establishment of genetic diversity and DNA fingerprinting analysis could be useful for genetic and genomic analysis and systematic utilization of the currently available genetic variation in upland cotton.

**Supplementary Materials:** The following supporting information can be downloaded at: https://www.mdpi.com/article/10.3390/agronomy12071513/s1, Table S1. The names, certification years, and pedigrees of the 79 early-maturing cotton accessions. Table S2. Fingerprinting data for the 79 early-maturing upland cotton accessions.

**Author Contributions:** H.C. designed the experiments; Z.K., H.C., M.K.I., A.G., L.G., C.X. and J.Z. conceived the experiments and analyzed the results; Z.K., H.C., C.X., L.G. and J.Z. carried out all computational analyses; Z.K., H.C., C.X., L.G., J.Z., W.W., B.W., H.H., Y.L. (Yujun Li) and Y.L. (Yuqiang Li) participated in part of the experiments; A.G., H.C., Z.K., S.S., A.T. and H.H. drafted the manuscript; H.H., S.K. and D.I. proofread and revised the manuscript; and H.C. and B.W. revised the manuscript. S.K. submitted the manuscript as corresponding author. All authors have read and agreed to the published version of the manuscript.

**Funding:** This program was financially sponsored by National Key R&D Program of China (2021YFE0101200), Hunan Natural Science Foundation Youth Fund (2020JJ5291), "Huxiang Young Talents Plan" Support Project of Hunan Province (2019RS2048), State Key Laboratory of Cotton Biology Open Fund (CB2021A14).

**Institutional Review Board Statement:** Not applicable.

**Informed Consent Statement:** Not applicable.

**Data Availability Statement:** Not applicable.

**Conflicts of Interest:** The authors declare no conflict of interest.

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
