# Peer review of "Use of SSR Markers for the Exploration of Genetic Diversity and DNA Finger-Printing in Early-Maturing Upland Cotton (Gossypium hirsutum L.) for Future Breeding Program"

_agronomy, doi:10.3390/agronomy12071513_

Round 1

Reviewer 1 Report

Assessing genetic diversity is an important task for any crop. SSR is the most suitable and widely used marker for this purpose. This study uses 71 SSR loci to characterize 79 early maturing cotton lots.

I suggest including the scientific name (Gossypium hirsutum L.) in the title of the article.

The manuscript is interesting and may be of interest, however, it has many drawbacks.
The introduction section is very short and should be written in more detail about the marker studies on cotton, highlighting the main findings of each study.
In the material and methods section, the methods used should be described in more detail so that anyone can replicate the experiments. In this context, it should be described:
How were the concentrations measured in the DNA extracts?
Was the length of SSR alleles (fragments) determined, and if so, by what method?
What was the composition of the PAGE gel? What buffer was used? What was the length and thickness of the gel?
Since cotton is an allotetraploid species, it is generally expected that the number of SSR alleles for a given genotype is usually up to 2 or 4 (if the locus is found in both ancestors). Has this been taken into account in the evaluation?
The manuscript does not include the detailed results (SSR profiles per genotype), I suggest that this is either corrected or included as supplementary material, or at least a link to the manuscript where this information is available.
In many places in the discussion section, you only describe previous research but do not compare the results with your own results. The introduction could be expanded by moving these sections there. In addition, the discussion section contains very valuable information. It might even be interesting to discuss whether specific alleles have been found that are unique to a particular geographic location or locations.

Author Response

Response Sheet

Subject:          Response to Comments

Title:   SSR Markers Explored the Genetic Diversity and DNA Fingerprinting in Early-maturing Upland Cotton for Future Breeding Program.

I, Shahbaz Khan, corresponding author of the manuscript, am enclosing herewith a revised manuscript entitled “SSR Markers Explored the Genetic Diversity and DNA Fingerprinting in Early-maturing Upland Cotton for Future Breeding Program” for publication in “Agronomy” after possible improvements. All the comments and suggestions are addressed accordingly and incorporated in the revised manuscript. Details of individual comments are given below.

General Comments

Assessing genetic diversity is an important task for any crop. SSR is the most suitable and widely used marker for this purpose. This study uses 71 SSR loci to characterize 79 early maturing cotton lots.

Response: Thank you so much for your time for reviewing our manuscript and providing comments and suggestions to improve the quality of manuscript.

Comment: I suggest including the scientific name (Gossypium hirsutum L.) in the title of the article.

Response: Suggestions are incorporated and highlighted.

Comment: The manuscript is interesting and may be of interest, however, it has many drawbacks. The introduction section is very short and should be written in more detail about the marker studies on cotton, highlighting the main findings of each study.

Response: Introduction section is improved accordingly. Suggestions are incorporated and highlighted.

Comment: In the material and methods section, the methods used should be described in more detail so that anyone can replicate the experiments. In this context, it should be described:
How were the concentrations measured in the DNA extracts?

Response: Suggestions are incorporated and highlighted.

Comment: Was the length of SSR alleles (fragments) determined, and if so, by what method?

Response: Suggestions are incorporated and highlighted.

Comment: What was the composition of the PAGE gel? What buffer was used? What was the length and thickness of the gel?

Response: Suggestions are incorporated and highlighted.

Comment: Since cotton is an allotetraploid species, it is generally expected that the number of SSR alleles for a given genotype is usually up to 2 or 4 (if the locus is found in both ancestors). Has this been taken into account in the evaluation?

Response: Number of alleles amplified by the 71 SSR primer pairs ranged from 1 to 6. Suggestions are incorporated and highlighted.

Comment: The manuscript does not include the detailed results (SSR profiles per genotype), I suggest that this is either corrected or included as supplementary material, or at least a link to the manuscript where this information is available.

Response: The information regarding DNA fingerprinting of each cotton cultivar is present as supplementary table at Table S2.

Comment: In many places in the discussion section, you only describe previous research but do not compare the results with your own results. The introduction could be expanded by moving these sections there.

Response: Suggestions regarding removing some parts of Discussion and adjusting in Introduction section has been incorporated and highlighted.

Comment: In addition, the discussion section contains very valuable information. It might even be interesting to discuss whether specific alleles have been found that are unique to a particular geographic location or locations.

Response: No primer pair was to be particularly specific for one geographical location. However, genetic diversity analysis on the basis of distinct regions of collection of studied cotton genotypes has been discussed in detail at Section 3.4.

A revised manuscript with track changes is attached for your kind consideration.

We say once again thanks to reviewer for the valuable and critical comments to improve the quality of manuscript.

Reviewer 2 Report

The manuscript coded agronomy-1762442, entitled "SSR Markers Explored the Genetic Diversity and DNA Fingerprinting in Early-maturing Upland Cotton for Future Breeding Program" is well written and fills the scope of the journal.   The only issue is the limited novelty of the study. Really a large number of articles have already been published on the application of SSR markers (or microsatellites) to cotton. The novelty of the study must therefore be better highlighted. It should be clarified, for instance, if the 79 accessions which are the object of this investigation have not been studied by other authors.   Other suggestions:   A table to illustrate the origin of each accession (China-different regions; Former Soviet Union; U.S.) should be added.   Other applications of SSRs, such as cultivar identification, or detection of GM genotypes, and more recent techniques than acrylamide gel and silver staining (which has a limited precision in estimating band size) should be acknowledged, such as HRM, either in the Introduction or in Results and discussion or even Conclusions, as further development. HRM analysis is highly suitable for the detection of single-base variants and small insertions or deletions. See, for example: https://doi.org/10.1016/j.foodchem.2014.02.111 and https://doi.org/10.1155/2015/496986   In the references, the botanical names should be italicized.  

Author Response

Response Sheet

Subject:          Response to Comments

Title:   SSR Markers Explored the Genetic Diversity and DNA Fingerprinting in Early-maturing Upland Cotton for Future Breeding Program.

I, Shahbaz Khan, corresponding author of the manuscript, am enclosing herewith a revised manuscript entitled “SSR Markers Explored the Genetic Diversity and DNA Fingerprinting in Early-maturing Upland Cotton for Future Breeding Program” for publication in “Agronomy” after possible improvements. All the comments and suggestions are addressed accordingly and incorporated in the revised manuscript. Details of individual comments are given below.

General Comments

The manuscript coded agronomy-1762442, entitled "SSR Markers Explored the Genetic Diversity and DNA Fingerprinting in Early-maturing Upland Cotton for Future Breeding Program" is well written and fills the scope of the journal.

Response: Thank you so much for your time for reviewing our manuscript and providing comments and suggestions to improve the quality of manuscript.

Comment: The only issue is the limited novelty of the study. Really a large number of articles have already been published on the application of SSR markers (or microsatellites) to cotton. The novelty of the study must therefore be better highlighted. It should be clarified, for instance, if the 79 accessions which are the object of this investigation have not been studied by other authors.

Response: Novelty statements has been improved and highlighted in last part of the Introduction section.

Comment: Other suggestions:   A table to illustrate the origin of each accession (China-different regions; Former Soviet Union; U.S.) should be added.

Response: Origin and pedigree information of all the 79 cotton accessions has been provided in supplementary material as Table S1.

Comment: Other applications of SSRs, such as cultivar identification, or detection of GM genotypes, and more recent techniques than acrylamide gel and silver staining (which has a limited precision in estimating band size) should be acknowledged, such as HRM, either in the Introduction or in Results and discussion or even Conclusions, as further development.

Response: Suggestions regrading HRM technique has been incorporated and highlighted in discussion section.

Comment: HRM analysis is highly suitable for the detection of single-base variants and small insertions or deletions. See, for example: https://doi.org/10.1016/j.foodchem.2014.02.111 and https://doi.org/10.1155/2015/496986.

Response: Thank you so much for providing the relevant material.

Comment: In the references, the botanical names should be italicized.

Response: Suggestions are incorporated and highlighted.

A revised manuscript with track changes is attached for your kind consideration.

We say once again thanks to reviewer for the valuable and critical comments to improve the quality of manuscript.

Round 2

Reviewer 1 Report

The revised version of the manuscript is suitable for publication.